# Assessing Doubts, Knowledge, and Service Appreciation among Pregnant Women Who Received the COVID-19 Vaccination in an Italian Research Hospital: A Cross-Sectional Study

**DOI:** 10.3390/vaccines11040812

**Published:** 2023-04-07

**Authors:** Stefania Bruno, Lorenza Nachira, Paola Arcaro, Fabio Pattavina, Enrica Campo, Chiara Cadeddu, Brigida Carducci, Antonio Lanzone, Gianfranco Damiani, Patrizia Laurenti

**Affiliations:** 1Women, Children and Public Health Sciences Department, Fondazione Policlinico Universitario A. Gemelli IRCCS, Largo Agostino Gemelli 8, 00168 Rome, Italy; stefania.bruno@unicatt.it (S.B.); fabio.pattavina@policlinicogemelli.it (F.P.); brigida.carducci@policlinicogemelli.it (B.C.); antonio.lanzone@policlinicogemelli.it (A.L.); patrizia.laurenti@policlinicogemelli.it (P.L.); 2Section of Hygiene, University Department of Life Sciences and Public Health, Università Cattolica del Sacro Cuore, Largo Francesco Vito 1, 00168 Rome, Italy; paola.arcaro01@icatt.it (P.A.); enrica.campo01@icatt.it (E.C.); chiara.cadeddu@unicatt.it (C.C.)

**Keywords:** pregnant women, vaccine hesitancy, COVID-19, occupational categories, vaccine, pandemic

## Abstract

The COVID-19 pandemic is considered one of the deadliest pandemics in history. Pregnant women are more susceptible to developing serious diseases during COVID-19 than their non-pregnant peers. Pregnant women often express doubt about accepting the vaccination, especially in regard to their security and safety. This study aims to investigate the appreciation of the vaccination offer, and if there are any determinants impacting vaccine hesitancy. A questionnaire was administered to a sample of pregnant women who had just received their immunization against COVID-19 at the vaccination service of a teaching hospital in Rome, from October 2021 to March 2022. A high appreciation of the vaccination services was found, both for the logistic organization and the healthcare personnel, with mean scores above 4 out of 5. The degree of pre-vaccinal doubt was low (41%) or medium (48%) for the largest part of the sample, while the degree of COVID-19 vaccine knowledge was high for 91% of the participants. Physicians were the most decisive information source for the vaccination choice. Our results highlighted that a supportive approach could increase appreciation and improve the setting of vaccinations. Healthcare professionals should aim for a more comprehensive and integrated role of all figures.

## 1. Introduction

Over the course of more than two years, the COVID-19 pandemic determined over 750 million confirmed cases and nearly 6.9 million deaths worldwide [1], making it one of the deadliest pandemics in history. The Italian scenario is currently characterized by 25.5 million confirmed cases and approximately 188,000 deaths [1]. A key mitigation strategy has been the rapid development of vaccines, which began in earnest in March 2020 and took place with active administration in December 2020 [2].

The COVID-19 vaccine was proven safe and effective at preventing serious illness and death by means of phase III clinical trials [3], excluding women who were pregnant or lactating. Despite the subsequent lack of evidence on these populations, the Centers for Disease Control and Prevention (CDC) and the American College of Obstetrics and Gynecology (ACOG) recommended that physicians advise pregnant and breastfeeding women to get the vaccination, especially those at high risk of infection-related complications [4,5]. This advice agreed with early research on the COVID-19 vaccination in lactating and pregnant women, which demonstrated that these vaccinations were safe, produced strong protection in recipients, and passed immunity to newborns through the placenta and breastmilk [6]. These encouraging results were especially significant given the increased danger that pregnant women experienced from COVID-19 infection.

In Italy, the Italian Obstetric Surveillance System (ItOSS) of the Italian National Institute of Health (ISS, Istituto Superiore di Sanità, Rome, Italy) recommended vaccination with mRNA vaccines in women at increased risk of contracting SARS-CoV-2 infection or developing severe COVID-19 disease through the publication of an interim document on 9 January 2021, later updated on 31 January 2021. On 22 September 2021, considering the growing real-world evidence on the effectiveness and safety of vaccination in pregnancy, the ISS recommended the extension of the COVID-19 vaccination offer to all pregnant women in their second and third trimesters [7].

Nevertheless, worries regarding vaccines and resistance to receiving the vaccination have been major obstacles to the large-scale immunization of pregnant and lactating women. Vaccine hesitancy, defined by the World Health Organization (WHO) as the reluctance or refusal to vaccinate despite the availability of vaccines [8], has been high in these populations of women since the mid-1990s, but has shown an increase in recent years. The novelty of the COVID-19 vaccine represents an additional determinant of the anti-vaccination sentiment, despite the high national and global burden of the COVID-19 pandemic [9]. Thus, vaccination uptake remains low among pregnant women worldwide [10].

In October 2021, the Fondazione Policlinico Universitario Agostino Gemelli IRCCS (FPG) in Rome set up a COVID-19 vaccination open day for pregnant women and created a specific pathway for them in its COVID-19 vaccination center. A cross-sectional study was conducted, aimed at (I) assessing the appreciation of the vaccination services and personnel among pregnant women who received the COVID-19 vaccination at the FPG, and its association with the sociodemographic characteristics; (II) assessing the women’s degree of knowledge and pre-vaccinal doubt about COVID-19 and its vaccines, and their association with the sociodemographic characteristics; (III) investigating which information sources were decisive for their vaccination choice.

## 2. Materials and Methods

### 2.1. Study Design

This cross-sectional study was conducted by administering a questionnaire to a convenience sample consisting of pregnant women who had started the primary COVID-19 vaccination course at the vaccination services offered by the FPG, from October 2021 to March 2022. As our study design was based on a convenience sampling approach, an estimate of the sample size was not necessary. The methodology used is in accordance with the most recent Guidelines for Observational Studies, STROBE (Strengthening the Reporting of Observational Studies in Epidemiology) [11]. 

### 2.2. Vaccination Services and Sample Recruitment

In light of the recent recommendations by the ISS, on 25 October 2021, the FPG organized a COVID-19 vaccination open day for women in the second and third trimesters of pregnancy who had not yet received any COVID-19 vaccine doses. The invitation for vaccination was addressed to pregnant women who accessed the outpatient clinic of the Obstetrics and Obstetric Pathology Unit of the FPG. In addition, the opportunity to receive the vaccination was extended to all pregnant women who expressed an interest. The administration of the first doses, in dedicated equipped rooms of the FPG, was preceded by an awareness-raising event where medical experts and a vaccinated pregnant woman advocated the importance and safety of the COVID-19 vaccination during pregnancy. 

Furthermore, in October, a specific pathway was dedicated to the COVID-19 vaccination of pregnant women at the FPG COVID-19 vaccination center to increase the enrollment of pregnant women. This pathway remained active over the following five months, until the closure of the vaccination center in March 2022. 

Pregnant women receiving their first dose of the vaccine during the open day, and later at the FPG vaccination center, were asked by the medical personnel to participate anonymously in a voluntary survey related to COVID-19. Those who consented were given an anonymous questionnaire to be completed during the observation period following the dose administration. Thus, the convenience sample was recruited.

### 2.3. Questionnaire and Data Collection 

Questionnaire face validity was detected by a panel of experts in epidemiology, statistics, public health, gynecology, and obstetrics. It consisted of three sections. 

The first section investigated sociodemographic and pregnancy-related characteristics through multiple-choice and open-ended questions. 

The second section explored the respondents’ satisfaction regarding the logistic organization and healthcare personnel at the vaccination services, and their knowledge regarding COVID-19 and pre-vaccinal doubt about its vaccines, through linear scale questions, with a Likert scale from 1 (totally disagree) to 5 (totally agree).

The third section investigated which information sources mostly influenced the respondents’ decision to receive the vaccination, through a multiple-choice question.

### 2.4. Data Analysis 

To assess the appreciation of the vaccination services, the mean value (M) and the standard deviation (SD) of the scores assigned by respondents to each related item were calculated.

To assess the degree of doubt, respondents were divided into three categories based on the sum of scores they attributed to each related item: low, medium, and high degree of doubt. A low degree of doubt was defined by sum values between 3 and 5, a medium degree ranged between 6 and 10, and a high degree between 11 and 15. For each category, the number and percentage of respondents were calculated.

The degree of knowledge was assessed using the same method as the degree of doubt.

To investigate which information sources were decisive for the participants’ vaccination choice, the answers were grouped into three categories: physicians (gynecologists, general practitioners, public health doctors, etc.), institutional sources (Ministry of Health, Local Health Authorities), and non-institutional sources (newspapers, blogs, forums, other websites). For each category, the number and percentage of respondents were calculated. Since multiple answers were allowed, the sum of the proportions did not necessarily reach 100%.

### 2.5. Statistical Analysis

Qualitative variables were expressed as absolute frequency and percentage. For quantitative variables, medians, and interquartile ranges (IQR), as well as means and standard deviations, were calculated. The overall coefficient of variation was calculated for two groups of questions: satisfaction regarding logistic organization and satisfaction regarding healthcare personnel. The Shapiro–Wilk test and Q–Q plots were employed to assess the variables’ distribution normality. A linear regression analysis was used to identify the correlation between logistic organization and appraisal of healthcare staff, with occupation and the level of education. A *t*-test and ANOVA analysis were performed to test the correlation between levels of doubt and knowledge with sociodemographic and pregnancy-related characteristics. All statistical tests were two-sided, a *p*-value < 0.05 was considered statistically significant. The analyses were performed using Stata 16.1 software (StataCorp LLC, College Station, TX, USA, 2019).

### 2.6. Ethical Statement

This study is compliant with the Local Ethical Committee Standards of the FPG. It was approved and registered (Prot. N◦ 38779/21 ID: 4557) and carried out in accordance with the Helsinki Declaration and EU Regulation 2016/679 (GDPR). For this kind of study, the Ethical Committee required the participant’s informed consent.

## 3. Results

Out of 102 women vaccinated, 100 women consented to participate in the study: a total of 22 answered the questionnaire during the open day on 25 October 2021, and 78 at the FPG COVID-19 vaccination center between October 2021 and March 2022. Their sociodemographic and pregnancy-related characteristics are shown in Table 1.

Concerning the appreciation of the vaccination service’s logistic organization, the Q–Q plot showed a normal distribution of the scores for each item. The median score was 5 for every questionnaire item. The mean scores for each questionnaire item had a range of 4.65–4.81 out of 5 (SD 0.54–0.87) among the whole sample, as shown in Table 2. The overall coefficient of variation was 10.0%. No association was found between the occupation and educational level.

Concerning the appreciation of the healthcare personnel, the Q–Q plot showed a normal distribution of scores for each item. The median score was 5 for every questionnaire item. The mean scores for each questionnaire item had a range of 4.31–4.80 out of 5 (SD 0.59–1.07) among the whole sample, as shown in Table 3. The overall coefficient of variation was 14.2%. No association was found between the occupation and educational level.

The degree of pre-vaccinal doubt was low in 41.0% of respondents, medium in 48.0%, and high in 11.0% (*p* = 0.07). No association was found with age, educational level, parity, and trimester of pregnancy (Table 4). Statistically significant differences were found among occupational categories, as respondents working in healthcare reported a higher prevalence of a low degree of doubt than among other occupations (*p* < 0.001), as shown in Table 4.

The degree of knowledge was low in 1.0% of respondents, medium in 8.0%, and high in 91.0% (*p* = 0.08). No association was found between sociodemographic and pregnancy-related characteristics (Table 5).

Regarding information sources, 92.0% of respondents included physicians among those decisive in their vaccination choice, of which 50.0% (46 women) chose no other source. Institutional sources were included by 33.0% of the sample, of which 3.0% (one woman) chose only this source. Finally, 35.0% of respondents included non-institutional sources, of which 17.1% (6 women) chose no other source. No association was found between information sources and individual characteristics, as reported in Table 6.

## 4. Discussion

Pregnant women are considered a high-risk population for severe COVID-19 and various pregnancy complications [12,13,14,15]. Despite the growing evidence on the safety and effectiveness of the COVID-19 vaccine in pregnancy, and recommendations from scientific societies and international public health institutions [5,16], coverage of the COVID-19 vaccination is still lower among pregnant women compared to the general population [17,18], and vaccine hesitancy is widespread among this population subgroup [19,20,21,22]. 

Based on this assumption, one of the purposes of this study was to evaluate the participants’ appreciation of an open day initiative and a specific pathway dedicated to the anti-COVID-19 vaccination of pregnant women. In addition, the study aimed to assess their degrees of knowledge and pre-vaccinal doubt about the vaccination, and the information sources deemed decisive for their vaccination choice.

The results indicated that the logistic organization and healthcare personnel of the vaccination services were highly appreciated, with mean scores above 4 out of 5. Linear regression allowed for us to analyze the relationship between variables, studying their direction and significance, and how they distributed the values of the two variables. Most participants exhibited a low (41%) or medium (48%) degree of pre-vaccinal doubt, while 91% had a high level of knowledge regarding COVID-19. Regarding information sources, physicians were the most decisive for their vaccination choice.

Our findings on sample characteristics including employment (75% of the women were employed), educational level (100% of women were high school or university graduates), and parity (96% were nulliparous) were consistent with previous research, indicating that specific socio-economic, educational, and obstetric factors play a significant role in vaccine acceptance. In a recent systematic review and meta-analysis, Bhattacharya et al. found that COVID-19 vaccine acceptance among pregnant women was highest among those who were employed, had at least twelve years of education, and were nulliparous [20]. Similarly, an Italian study on adherence to recommended vaccinations during pregnancy (influenza and pertussis) demonstrated a significant association with vaccination uptake, higher educational level, and employment status (employed, part-time, or full-time) [23]. Another study conducted in France also found that acceptance of the influenza vaccination among pregnant women was most prevalent among those with a high educational level and low parity (mainly nulliparous), and who are more attentive to preventive measures during pregnancy than multiparous women [24]. In contrast with the prevailing literature, Tao et al. reported a higher COVID-19 vaccine acceptance rate among pregnant women with lower educational levels in China, which may be due to differences in the determinants of vaccine acceptance in their population [25].

The level of appreciation expressed by the sample for all aspects of the logistical organization and healthcare personnel of vaccination services was notably high. These findings suggest the relevance of providing support to pregnant women even during the vaccination, reassuring them with scientific evidence, and reinforcing their beliefs through effective communication in a dedicated pathway. As highlighted by Carbone et al., pregnancy is an emotional phase in women’s lives that can lead to anxiety [26], which may be further exacerbated by the COVID-19 pandemic [27]. Therefore, efforts targeted towards this vulnerable population could significantly enhance the patient-perceived quality of services and increase patient trust and loyalty towards healthcare providers and the healthcare system [28]. Strengthening confidence through the adoption of effective strategies is essential in reducing vaccine hesitancy and, consequently, improving the success of any immunization plan [29,30]. 

Along with trust, knowledge about COVID-19 and concerns about vaccine safety and effectiveness are the main determinants of COVID-19 vaccine hesitancy [31]. People who lack adequate knowledge about the disease, its transmission, and the vaccine development and approval process are less likely to receive the COVID-19 vaccine, while misinformation, rumors, and fear of adverse events can also fuel hesitancy [19,32,33,34]. As proof of that, Regan et al. reported that most pregnant women who received or planned to receive the COVID-19 vaccine believed that it was safe, that its advantages outweighed its risks, and that it would protect them from the disease [17]. In agreement with the existing literature, our results showed a high degree of knowledge (91% of the survey respondents) and a low-to-medium degree of pre-vaccinal doubt (low in 41% and medium in 48%) among women who had just received the vaccine, with a higher prevalence of a low degree of doubt among healthcare workers (HCWs) compared to other occupational categories (*p* < 0.001). 

It is worth noting that other factors, including personal experiences, cultural beliefs, and interactions with the healthcare system, can influence a person’s perception of the vaccine, determining uncertainty and anxiety and fueling doubts. This may contribute to explaining why HCWs may also experience vaccine hesitancy, despite having advanced education in the sciences and clinical expertise [35]. On this topic, a scoping review conducted by Biswas et al. reported that COVID-19 vaccine hesitancy among HCWs ranged from 4.3 to 72% worldwide, mainly caused by concerns about vaccine safety, efficacy, and potential adverse effects [36]. Similarly, a study conducted by Bianchi et al. estimated a vaccine hesitancy rate of 13.1% among HCWs in Italy (ranging from 18.2% to 8.9% before and during the vaccination campaign, respectively), with a lack of information about the vaccination, doubts about vaccine safety, and fear of side effects being the main determinants [37]. Another Italian study showed that 17.0% of HCWs were hesitant toward vaccination in general, 32.3% were hesitant toward the COVID-19 vaccination, while 18.8% were classified as refusing obligations, regarding COVID-19-related injunctive measures [38].

Vaccine hesitancy among HCWs is a serious concern because of its implications for patient safety and the healthcare system, given HCWs’ crucial role in promoting and advocating for vaccines [35]. In this regard, our results showed that most study participants (92%) included physicians among those who were decisive for their vaccination choice, half of which listed physicians as their only answer. These findings aligned with those of previous studies, which highlighted the influential role of physicians in the vaccination decision-making process for pregnant women. For example, a cross-sectional survey of US pregnant women showed that respondents who were recommended for COVID-19 vaccination by their healthcare provider were 52% more likely to accept the vaccine, reporting fewer concerns about vaccine safety and greater confidence in its effectiveness, compared to respondents with no provider recommendation [17]. Similarly, a Korean study demonstrated that patient education about maternal COVID-19 vaccination on the behalf of obstetrics and gynecology doctors is pivotal in increasing vaccination coverage [39]. Furthermore, a report from the CDC found that healthcare provider recommendations were associated with a higher likelihood of COVID-19 vaccination (77.6% vs. 61.9% of adults who did not receive a recommendation) [40]. Beyond COVID-19, the role of healthcare professionals was consistently identified by the scientific literature as a significant determinant of the uptake of other vaccines recommended for pregnancy, resulting in high vaccination coverage [41,42,43].

One of the strengths of our study is that it demonstrates how a supportive approach can increase appreciation and improve the vaccination environment. However, it is worth noting that the population we studied was already inclined towards vaccination, as participants were recruited at the vaccination center; therefore, these results are not representative of the pregnant women who decided not to receive the vaccination. We believe that our study may provide valuable insights into the experiences and perspectives of pregnant women who accepted vaccination for an emerging infectious disease, which could inform strategies to improve vaccine uptake among pregnant women who may be on the fence about getting vaccinated. Additionally, the study results can be used to identify areas where healthcare providers can improve the quality of care provided to pregnant women who have decided to get vaccinated. The convenience sampling approach represents a limitation of our study, as its results may not be used to make inferences about the attitudes and behaviors of the broader population. Moreover, including a greater sample of pregnant women could highlight significant differences between some subgroups. Another limitation is the lack of investigation into clinical conditions that could influence the degree of doubt.

In the future, a more comprehensive and integrated model involving all healthcare professionals, besides obstetrics and gynecology doctors, could be implemented to promote all vaccinations recommended during pregnancy. The life course approach could be adopted to further increase trust in immunization, emphasizing the importance of vaccinations at all ages.

## 5. Conclusions

Vaccination against SARS-CoV-2 during pregnancy is an act of primary prevention effective for both mothers and newborns who are at higher risk of a severe outcome of the disease. However, improving immunization rates requires a continuous, integrated, multi-level, and synergic approach that addresses the unique needs and concerns of these vulnerable populations [22]. Policymakers and healthcare professionals should implement tailored strategies and interventions to improve vaccine literacy [21], reduce concerns about vaccine safety and effectiveness, and ensure access to accurate and up-to-date information. These measures can help to improve immunization rates, overcome resistance, and protect—with professional competence—pregnant women and their newborns from the severe consequences of COVID-19 and other vaccine-preventable diseases.

## Figures and Tables

**Table 1 vaccines-11-00812-t001:** Sociodemographic and pregnancy-related characteristics (N = 100).

Variables	n (%)
**Citizenship**	
Italian	97 (97)
Other	3 (3)
**Marital status**	
Married or living with a partner	98 (98)
Single, widowed, or divorced/separated	2 (2)
**Educational level**	
Graduate	82 (82)
High school	18 (18)
**Occupation**	
Employee (excluding healthcare workers)	32 (32)
Self-employed or managerial role	20 (20)
Healthcare worker	24 (24)
Job searching, unemployed, or student	24 (24)
**Parity**	
0	96 (96)
1 or more	4 (4)
**Trimester of Pregnancy**	
Second Trimester	47 (47)
Third Trimester	53 (53)
**Age**	
20–27 years old	10 (10)
28–35 years old	60 (60)
36–44 years old	30 (30)

**Table 2 vaccines-11-00812-t002:** Appreciation of the logistic organization of the vaccination services, related to occupation and educational level (N = 100).

Items	Occupation					Educational Level			Overall M (SD)
	Employee (Excluding Healthcare Workers) M (SD)	Self-Employed or Managerial Role M (SD)	Healthcare Worker M (SD)	Job Searching, Unoccupied or Student M (SD)	*p*-Value	Graduate M (SD)	High School M (SD)	*p*-Value	
Acceptability of the waiting time before vaccination	4.8 (0.64)	4.68 (0.58)	4.83 (0.48)	4.71 (0.86)	0.87	4.73 (0.71)	4.94 (0.23)	0.36	4.77 (0.65)
Convenience of using the vaccination booking system	4.75 (0.80)	4.95 (0.22)	4.79 (0.51)	4.79 (0.66)	0.96	4.78 (0.67)	4.94 (0.23)	0.69	4.81 (0.61)
Clarity of information about the place and time of vaccination	4.56 (1.1)	4.65 (0.67)	4.71 (0.91)	4.71 (0.69)	0.86	4.67 (0.86)	4.55 (0.92)	0.41	4.65 (0.87)
Comfort of the environments	4.81 (0.47)	4.65 (0.74)	5 (0)	4.75 (0.85)	0.96	4.79 (0.64)	4.89 (0.32)	0.75	4.81 (0.6)
Hygiene of the environments	4.75 (0.51)	4.7 (0.57)	4.92 (0.41)	4.75 (0.85)	0.95	4.79 (0.60)	4.72 (0.57)	0.48	4.78 (0.6)
Guarantee of privacy in the environments	4.62 (0.66)	4.75 (0.72)	4.96 (0.20)	4.58 (0.97)	0.53	4.67 (0.75)	4.94 (0.23)	0.25	4.72 (0.7)
Overall satisfaction with the organization	4.84 (0.37)	4.85 (0.37)	4.91 (0.28)	4.67 (0.92)	0.99	4.79 (0.58)	4.94 (0.23)	0.19	4.77 (0.54)

**Table 3 vaccines-11-00812-t003:** Appreciation of the healthcare personnel at the vaccination services, related to occupation and educational level (N = 100).

Items	Occupation					Educational Level			Overall M (SD)
	Employee (Excluding Healthcare Workers) M (SD)	Self-Employed or Managerial Role M (SD)	Healthcare Worker M (SD)	Job Searching, Unoccupied or Student M (SD)	*p*-Value	Graduate M (SD)	High School M (SD)	*p*-Value	
Thoroughness and clarity of the information received from the medical staff about the benefits and risks of vaccination	4.03 (1.14)	4.4 (1.14)	4.62 (0.92)	4.29 (1.00)	0.56	4.35 (1.05)	4.11 (1.18)	0.47	4.31 (1.07)
Professionality and competence of the medical staff	4.43 (0.84)	4.75 (0.55)	4.71 (0.62)	4.62 (0.87)	0.67	4.61 (0.75)	4.61 (0.78)	0.58	4.61 (0.75)
Professionality and competence of the nursing staff	4.65 (0.65)	4.8 (0.52)	4.87 (0.34)	4.75 (0.85)	0.29	4.76 (0.64)	4.77 (0.56)	0.81	4.76 (0.62)
Kindness and willingness to listen to the medical staff	4.59 (0.66)	4.85 (0.49)	5 (0)	4.67 (0.87)	0.15	4.74 (0.66)	4.83 (0.38)	0.48	4.76 (0.62)
Kindness and willingness to listen to the nursing staff	4.66 (0.60)	4.85 (0.49)	5(0)	4.75 (0.85)	0.53	4.78 (0.63)	4.89 (0.32)	0.76	4.8 (0.59)

**Table 4 vaccines-11-00812-t004:** Classification of the participants based on their degree of pre-vaccinal doubt (N = 100).

Sociodemographic and Pregnancy-Related Characteristics	Degree of Doubt			
	Low n (%)	Medium n (%)	High n (%)	*p*-Value
**Occupation**				
Employee (excluding healthcare workers)	8 (25.00)	23 (71.88)	1 (3.13)	<0.001
Self-employed or managerial role	8 (40.00)	8 (40.00)	4 (20.00)	
Healthcare worker	19 (79.17)	4 (16.67)	1 (4.17)	
Job searching, unemployed, or student	6 (25.00)	13 (54.17)	5 (20.83)	
**Age**				
20–27 years old	4 (44.44)	4 (44.44)	1 (11.11)	0.98
28–35 years old	24 (40.00)	30 (50.00)	6 (10.00)	
36–44 years old	12 (40.00)	14 (46.67)	4 (13.33)	
**Educational Level**				
Graduate	31 (37.80)	41 (50.00)	10 (12.20)	0.34
High school	10 (55.56)	7 (38.89)	1 (5.56)	
**Parity**				
0	2 (50.00)	1 (25.00)	1 (25.00)	0.53
1 or more	39 (40.63)	47 (48.96)	10 (10.42)	
**Trimester of Pregnancy**				
Second	19 (41.30)	22 (47.83)	5 (10.87)	0.99
Third	22 (41.51)	25 (47.17)	6 (11.32)	
**Overall**	41 (41)	48 (48)	11 (11)	0.07

**Table 5 vaccines-11-00812-t005:** Classification of the participants based on their degree of knowledge (N = 100).

Sociodemographic and Pregnancy-Related Characteristics	Degree of Knowledge			
	Low n (%)	Medium n (%)	High n (%)	*p*-Value
**Occupation**				
Employee (excluding healthcare workers)	0	2 (11.1)	16 (88.9)	0.53
Self-employed or managerial role	0	4 (12.5)	28 (87.5)	
Healthcare worker	0	1 (4.17)	23 (95.8)	
Job searching, unemployed, or student	1 (4.17)	1 (4.17)	22 (91.7)	
**Age**				
20–27 years old	0 (0)	0 (0)	9 (100)	0.58
28–35 years old	1 (1.67)	6 (10.0)	53 (88.33)	
36–44 years old	0 (0)	1 (3.33)	29 (96.67)	
**Educational Level**				
Graduate	1 (1.22)	6 (7.32)	75 (91.5)	0.78
High school	0	2 (11.11)	16 (88.89)	
**Parity**				
0	0	0	4 (100)	0.81
1 or more	1 (1.04)	8 (8.33)	87 (90.63)	
**Trimester of Pregnancy**				
Second Trimester	1 (2.17)	4 (7.55)	41 (89.1)	0.54
Third Trimester	0	4 (7.55)	49 (94.45)	
**Overall**	1 (1)	8 (8)	91 (91)	0.08

**Table 6 vaccines-11-00812-t006:** Information sources considered decisive for the vaccination choice (N = 100).

Individual Characteristics	Information Sources			
	Physicians n (%)	Institutional Sources n (%)	Non-Institutional Sources n (%)	*p*-Value
**Occupation**				
Employee (excluding healthcare workers)	29 (44.62)	24 (36.92)	12 (18.46)	0.30
Self-employed or managerial role	19 (57.58)	8 (24.24)	6 (18.18)	
Healthcare worker	20 (52.64)	9 (23.68)	9 (23.68)	
Job searching, unemployed, or student	24 (60.00)	8 (20.00)	8 (20.00)	
**Age**				
20–27 years old	8 (50.00)	4 (25.00)	4 (25.00)	0.25
28–35 years old	54 (58.06)	20 (21.51)	19 (20.43)	
36–44 years old	29 (58.00)	9 (18.00)	12 (24.00)	
**Educational Level**				
Graduate	75 (57.69)	28 (31.54)	27 (20.77)	0.40
High school	17 (56.67)	5 (16.67)	8 (26.66)	
**Degree of doubt**				
Low	36 (54.54)	15 (22.73)	15 (22.73)	0.38
Medium	45 (59.21)	14 (18.42)	17 (22.37)	
High	11 (61.11)	4 (22.22)	3 (16.67)	
**Degree of knowledge**				
Low	1 (33)	1 (33)	1 (33)	0.80
Medium	6 (54.54)	2 (18.18)	3 (27.28)	
High	85 (57.4)	31 (20.95)	32 (21.62)	
**Overall**	92	33	35	

## Data Availability

Not applicable.

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
