# Peer review of "Assessing Doubts, Knowledge, and Service Appreciation among Pregnant Women Who Received the COVID-19 Vaccination in an Italian Research Hospital: A Cross-Sectional Study"

_vaccines, 2023, doi:10.3390/vaccines11040812_

Round 1

Reviewer 1 Report

Thank you very much for giving me the valuable opportunity to review the paper. In this paper, the authors aimed to investigate the appreciation of the vaccination offer, and if there are any determinants impacting on vaccine hesitancy. They found a high appreciation of the vaccination services both for the logistic organization and the healthcare personnel with mean scores above 4 out of 5. The degree of pre-vaccinal doubt was low (41%) or medium (48%) for the largest part of the sample, while the degree of COVID-19 vaccine knowledge was high for 91% of the participants. Physicians were the most decisive source for the vaccination choice. This study is interesting, but there are some limitations: (1) The sample size was 100 pregnant women in this study, which was relative small, the authors should provide the estimate of sample size and power of test; (2) Other determinants might impact the vaccine hesitancy among pregnant women, such as gestational complications, certain comorbidities, and history of drug or vaccine allergy, but the information was missing presently.

Author Response

Thank you for your interest in our manuscript. We appreciate your feedback and will do our best to address your concerns. Please see the attachment for our response.

Response to Reviewer 1 Comments

Point 1: The sample size was 100 pregnant women in this study, which was relative small, the authors should provide the estimate of sample size and power of test.

Response 1: Thank you for your suggestion, we understand your concerns about sample size. However, we are convinced that an estimate of sample size and power of the test is not indicated, since ours is a convenience sample. We understand that our manuscript could be vague on this point, therefore we added clarification at lines 88-89 of the track changes revision.

Point 2: Other determinants might impact the vaccine hesitancy among pregnant women, such as gestational complications, certain comorbidities, and history of drug or vaccine allergy, but the information was missing presently.

Response 2: Thank you for bringing this to our attention. We agree that vaccine hesitancy is a complex issue, however we did not address this topic in our study: indeed, we recruited pregnant women at the time of vaccination, because one of our objectives was to understand which doubts they had overcome in order to get vaccinated; therefore, by definition, we cannot refer to their doubts as “vaccine hesitancy”. Regarding clinical conditions (such as gestational complications, comorbidities, and allergological history), we did not investigate these aspects in our questionnaire, because our objective was to assess the degree of pre-vaccinal doubt and explore the correlation with socio-demographic factors. We recognize other studies may be needed to explore other determinants and acknowledge it as a limitation in the discussion (lines 309-310).

Reviewer 2 Report

Thank you for asking me to review the paper “Assessing Doubts, Knowledge, and Service Appreciation among Pregnant Women who Received COVID-19 Vaccination in an Italian Research Hospital: a Cross-Sectional Study” by Bruno et al. 

As whole, the paper has a number of flaws that need to be addressed before considering it for publication. 

In Intro and Methods, authors mentioned the FPG vaccination open day, but it is not clear the importance of this point in study context and conduction. Were Women only enrolled during this open day? If is, the study can only be considered as a pilot study. And limited as a real-world study. 

Line 55: The first interim document was published on 9 January 2021, and updated on 31 January 2021, after the AstraZeneca vaccine was licensed. On 22 September 2021, the ISS updated the recommendations in its original document with the aim of helping healthcare professionals and pregnant and breastfeeding mothers make informed decisions during the COVID-19 pandemic. See: https://www.epicentro.iss.it/vaccini/pdf/indicazioni-vaccini-covid-gravidanza-allattamento.pdf

Sample size is very small. Although many scholars concur that a sample size of 100 is the minimum for having for meaningful results, I have doubt that in this case it work, particularly owing to the non-probability sampling strategy (line 100-101). Authors might want to justify the number of enrolled women, also comparing it to the total number of pregnant women who visited the vaccination center in the study period.  

From a methodological standpoint, there’s a Damocles’ sword hanging on the risk of ascertainment bias. When authors analysis customer (I.e., pregnant women) satisfaction , they were interviewing only subjects who did accede a Vaccination Center, thus biased towards a positive attitude of vaccine utility and vaccination uptake, leaving outside all the rest of community who decides to not vaccinate. This clearly explains low degree of “pre-vaccination doubt”, and I hardly understand the added value of the issue. Again on this point, the surveyed women were sampled during booster dose administration (after October 2021), thus enrolling individual who were already prone to receiving primary COVID-19 vaccination (even before to get pregnant). Authors cursorily acknowledged this limitation, while a more in-depth analysis/comment is needed, highlighting caution when interpreting their findings. Similarly, Comparisons with other similar studies are needed. 

For result presentation, with most item measured as scale data we cannot use the mean as a measure of central tendency as it has no meaning i.e. what is the average of Strongly agree and disagree? The most appropriate measure of is the mode the most frequent responses, or the median.

For result presentation, with Likert scale data we cannot use the mean as a measure of central tendency as it has no meaning i.e. what is the average of “Totally agree and disagree”? The most appropriate measure of is the mode the most frequent responses, or the median.

Author Response

Thank you for your interest in our manuscript. We appreciate your feedback and will do our best to address your concerns. Please see the attachment for our response.

Response to Reviewer 2 Comments

Point 1: In Intro and Methods, authors mentioned the FPG vaccination open day, but it is not clear the importance of this point in study context and conduction. Were Women only enrolled during this open day? If is, the study can only be considered as a pilot study. And limited as a real-world study.

Response 1: Thank you for bringing this to our attention. The FPG vaccination open day was a significant event in increasing awareness about the official stance of the Associations of Gynecologists and Obstetrics as well as international and national health institutions regarding COVID-19 vaccination during pregnancy. As detailed in the Results section (lines 161-163 of the track changes revision), 22 women were enrolled in the study during the open day and an additional 78 women at the FPG COVID-19 vaccination center over the following five months. Given the number of participants and the length of the study period, we agree that considering the study as a pilot may not be appropriate and we have provided additional details in the manuscript (lines 93-114) to clarify this point.

Point 2: Line 55: The first interim document was published on 9 January 2021, and updated on 31 January 2021, after the AstraZeneca vaccine was licensed. On 22 September 2021, the ISS updated the recommendations in its original document with the aim of helping healthcare professionals and pregnant and breastfeeding mothers make informed decisions during the COVID-19 pandemic. See: https://www.epicentro.iss.it/vaccini/pdf/indicazioni-vaccini-covid-gravidanza-allattamento.pdf

Response 2: Thank you for your feedback, which allowed us to improve consistency with the chronology of published documents. To enhance clarity, we included the complete sequence of documents issued by the ISS in our manuscript at lines 55-65 and changed reference no. 7 accordingly.

Point 3: Sample size is very small. Although many scholars concur that a sample size of 100 is the minimum for having for meaningful results, I have doubt that in this case it work, particularly owing to the non-probability sampling strategy (line 100-101). Authors might want to justify the number of enrolled women, also comparing it to the total number of pregnant women who visited the vaccination center in the study period.

Response 3: Thank you for your comment. We understand your concerns about the sample size, but since we used a convenience sampling approach, we were able to enroll in the study 100 out of the 102 pregnant women who were vaccinated at the open day and FPG vaccination center during the study period, which lasted, until the closure of the center itself in March 2022. Following your suggestion, we have included details about the total number at line 161 of our manuscript; moreover, we clarified at line 109 that the sample recruiting ended when the vaccination center closed..

Point 4: From a methodological standpoint, there’s a Damocles’ sword hanging on the risk of ascertainment bias. When authors analysis customer (I.e., pregnant women) satisfaction , they were interviewing only subjects who did accede a Vaccination Center, thus biased towards a positive attitude of vaccine utility and vaccination uptake, leaving outside all the rest of community who decides to not vaccinate. This clearly explains low degree of “pre-vaccination doubt”, and I hardly understand the added value of the issue. Again on this point, the surveyed women were sampled during booster dose administration (after October 2021), thus enrolling individual who were already prone to receiving primary COVID-19 vaccination (even before to get pregnant). Authors cursorily acknowledged this limitation, while a more in-depth analysis/comment is needed, highlighting caution when interpreting their findings. Similarly, Comparisons with other similar studies are needed.

Response 4: Thank you for your comments. We appreciate your concerns regarding the potential for ascertainment bias in our study. However, we would like to emphasize that the aim of our study was not to draw inferences about the entire population of pregnant women. Instead,  our focus was on investigating knowledge, doubts levels, reliance on information sources, and service satisfaction of a specific sample, consisting of pregnant women who decided to get vaccinated at FPG vaccination services.

We agree that our study population is not representative of all pregnant women, and we have added this clarification in our manuscript (lines 296-299). However, we believe the added value of our study is to provide relevant insights into the experiences and perspectives of pregnant women who accepted vaccination for an emerging infectious disease, that could inform strategies to improve vaccine uptake among pregnant women who may be on the fence about getting vaccinated. Additionally, the study results can be used to identify areas where healthcare providers can improve the quality of care provided to pregnant women who have decided to get vaccinated (as also stated in the discussion section at lines 299-305).

Regarding the vaccine dose administered, we enrolled only pregnant women getting their first dose of the vaccine (lines 86-88, 110-112). About the need of comparisons with other studies, we had already included comparisons with similar studies in the discussion section, however there aren’t many studies which investigate vaccinated pregnant women’s attitudes about COVID-19 vaccination, nor their appreciation of the vaccination services.

Point 5: For result presentation, with most item measured as scale data we cannot use the mean as a measure of central tendency as it has no meaning i.e. what is the average of Strongly agree and disagree? The most appropriate measure of is the mode the most frequent responses, or the median.

Point 6: For result presentation, with Likert scale data we cannot use the mean as a measure of central tendency as it has no meaning i.e. what is the average of “Totally agree and disagree”? The most appropriate measure of is the mode the most frequent responses, or the median.

Responses 5-6: Thank you for your feedback. We understand your concern about the use of mean as a measure of central tendency for Likert scale data. In our study, we used mean values because the responses had a normal distribution, and we wanted to capture even minor variations in the participants' responses. As you can see in the following tables reporting the median values and interquartile ranges (IQR), median values are very similar to each other, therefore they don’t allow for a more detailed analysis of the data. However, we acknowledge that we were not clear about our choices, therefore we added clarifications in the manuscript (lines 145-148).

Table 2. Appreciation of the logistic organization of the vaccination services, related to occupation and educational level (N=100)

Items

Occupation

Educational level

Overall

Median (IQR)

Employee (excluding healthcare workers

Median (IQR)

Self-employed or managerial role

Median (IQR)

Healthcare worker

Median (IQR)

Job searching, unoccupied or student

Median (IQR)

p-value

Graduate

Median (IQR)

High school

Median (IQR)

p-value

Acceptability of the waiting time before vaccination

5.0 (5.0-5.0)

5.0 (4.0-5.0)

5.0 (5.0-5.0)

5.0 (5.0-5.0)

0.87

5.0 (5.0-5.0)

5.0 (5.0-5.0)

0.36

5.0 (5.0-5.0)

Convenience of use of the vaccination booking system

5.0 (5.0-5.0)

5.0 (5.0-5.0)

5.0 (5.0-5.0)

5.0 (5.0-5.0)

0.96

5.0 (5.0-5.0)

5.0 (5.0-5.0)

0.69

5.0 (5.0-5.0)

Clarity of information about the place and time of vaccination

5.0 (5.0-5.0)

5.0 (4.5-5.0)

5.0 (5.0-5.0)

5.0 (5.0-5.0)

0.86

5.0 (5.0-5.0)

5.0 (5.0-5.0)

0.41

5.0 (5.0-5.0)

Comfort of the environments

5.0 (5.0-5.0)

5.0 (4.5-5.0)

5.0 (5.0-5.0)

5.0 (5.0-5.0)

0.96

5.0 (5.0-5.0)

5.0 (5.0-5.0)

0.75

5.0 (5.0-5.0)

Hygiene of the environments

5.0 (5.0-5.0)

5.0 (4.5-5.0)

5.0 (5.0-5.0)

5.0 (5.0-5.0)

0.95

5.0 (5.0-5.0)

5.0 (5.0-5.0)

0.48

5.0 (5.0-5.0)

Guarantee of privacy of the environments

5.0 (4.0-5.0)

5.0 (5.0-5.0)

5.0 (5.0-5.0)

5.0 (5.0-5.0)

0.53

5.0 (5.0-5.0)

5.0 (5.0-5.0)

0.25

5.0 (5.0-5.0)

Overall satisfaction with the organization

5.0 (5.0-5.0)

5.0 (5.0-5.0)

5.0 (5.0-5.0)

5.0 (5.0-5.0)

0.99

5.0 (5.0-5.0)

5.0 (5.0-5.0)

0.19

5.0 (5.0-5.0)

Table 3. Appreciation of the healthcare personnel at the vaccination services, related to occupation and educational level (N=100)

Items

Occupation

Educational level

Overall

M (SD)

Employee (excluding healthcare workers)

Median (IQR)

Self-employed or managerial role

Median (IQR)

Healthcare worker

Median (IQR)

Job searching, unoccupied or student

Median (IQR)

p-value

Graduate

Median (IQR)

High school or lower

Median (IQR)

p-value

Thoroughness and clarity of the information received from the medical staff about the benefits and risks of vaccination 

5.0 (3.0-5.0)

5.0 (4.0-5.0)

5.0 (5.0-5.0)

5.0 (4.0-5.0)

0.56

5.0 (4.0-5.0)

5.0 (3.0-5.0)

0.47

5.0 (4.0-5.0)

Professionality and competence of the medical staff

5.0 (4.0-5.0)

5.0 (5.0-5.0)

5.0 (5.0-5.0)

5.0 (4.5-5.0)

0.67

5.0 (4.0-5.0)

5.0 (4.0-5.0)

0.58

5.0 (4.0-5.0)

Professionality and competence of the nursing staff

5.0 (4.5-5.0)

5.0 (5.0-5.0)

5.0 (5.0-5.0)

5.0 (5.0-5.0)

0.29

5.0 (5.0-5.0)

5.0 (5.0-5.0)

0.81

5.0 (5.0-5.0)

Kindness and willingness to listen of the medical staff

5.0 (4.0-5.0)

5.0 (5.0-5.0)

5.0 (5.0-5.0)

5.0 (5.0-5.0)

0.15

5.0 (5.0-5.0)

5.0 (5.0-5.0)

0.48

5.0 (5.0-5.0)

Kindness and willingness to listen of the nursing staff

5.0 (4.0-5.0)

5.0 (5.0-5.0)

5.0 (5.0-5.0)

5.0 (5.0-5.0)

0.53

5.0 (5.0-5.0)

5.0 (5.0-5.0)

0.76

5.0 (5.0-5.0)

Reviewer 3 Report

1.       A linear regression analysis was used to identify the correlation between logistic organization and appraisal of healthcare staff with the level of occupation and education. Discuss, the authors should explain why the linear regression work quite well.

2.       The authors should sketch the data to show the scattering-shaped.

3.       Non-Parametric plots (kernel density – Box – strip – violin – “quantile- quantile”) should be sketched to discuss the behavior of data.

4.       The authors applied ANOVA test. Did the authors test the normality and homogeneity properties before analyzing data?.

5.       The authors used a linear regression model, the residuals and reliability of model should be discussed.

6.        Residuals plots should be added.

7.       The coefficient of variation (CV) should be calculated between groups to discuss which is more dispersion?.

Author Response

Thank you for your interest in our manuscript. We appreciate your feedback and will do our best to address your concerns. Please see the attachment for our response.

Round 2

Reviewer 1 Report

This paper can be accepted in present form.

Reviewer 2 Report

Authors addressed all the raised concerns 

Reviewer 3 Report

All comments have been answered.